# Activation of KCNQ4 as a Therapeutic Strategy to Treat Hearing Loss

**DOI:** 10.3390/ijms22052510

**Published:** 2021-03-02

**Authors:** John Hoon Rim, Jae Young Choi, Jinsei Jung, Heon Yung Gee

**Affiliations:** 1Department of Pharmacology, Graduate School of Medical Science, Brain Korea 21 Project, Yonsei University College of Medicine, Seoul 03722, Korea; JOHNHOON1@yuhs.ac; 2Department of Otorhinolaryngology, Graduate School of Medical Science, Brain Korea 21 Project, Yonsei University College of Medicine, Seoul 03722, Korea; JYCHOI@yuhs.ac

**Keywords:** potassium voltage-gated channel subfamily q member 4, potassium, hearing loss, nonsyndromic hearing loss, KCNQ4 activator

## Abstract

Potassium voltage-gated channel subfamily q member 4 (KCNQ4) is a voltage-gated potassium channel that plays essential roles in maintaining ion homeostasis and regulating hair cell membrane potential. Reduction of the activity of the KCNQ4 channel owing to genetic mutations is responsible for nonsyndromic hearing loss, a typically late-onset, initially high-frequency loss progressing over time. In addition, variants of KCNQ4 have also been associated with noise-induced hearing loss and age-related hearing loss. Therefore, the discovery of small compounds activating or potentiating KCNQ4 is an important strategy for the curative treatment of hearing loss. In this review, we updated the current concept of the physiological role of KCNQ4 in the inner ear and the pathologic mechanism underlying the role of KCNQ4 variants with regard to hearing loss. Finally, we focused on currently developed KCNQ4 activators and their pros and cons, paving the way for the future development of specific KCNQ4 activators as a remedy for hearing loss.

## 1. Introduction

Hearing impairment, the most common sensory deficit in humans, affects 466 million people (over 6% of the world’s population) according to the World Health Organization (WHO) (https://www.who.int/news-room/fact-sheets/detail/deafness-and-hearing-loss) (accessed on 20 January 2021) [1,2]. Approximately 1 in 500–1000 individuals suffer from congenital hearing loss, of which approximately 50% are known to be caused by genetic mutations [3,4]. The prevalence of hearing loss has been reported to double with every 10-y increase in age, with almost two-thirds of individuals over the age of 70 y having a hearing impairment associated with sounds ≥ 25 dB [2]. The main causes of adult-onset hearing loss are noise exposure, aging, genetic mutations, exposure to therapeutic drugs that have ototoxic side-effects, viruses, or ototoxic drugs or chemicals, resulting in damage to the auditory hair cells and neurons [1,2]. As the number of aging adults is increasing globally, hearing loss poses a high economic burden, with an estimated cost of $750 billion annually [5]. Despite this, all available treatment options for hearing loss to date are limited to hearing devices, such as hearing aids and cochlear implants. Medical treatments are both lacking and required.

As with almost all other sensory transduction systems, hearing involves the modulation of potassium (K^+^) channels at an early stage of the process of turning mechanical sound into electrical signals. Potassium voltage-gated channel subfamily q member 4 (KCNQ4) is known to play an essential role in the auditory function of the inner ear, contributing to potassium recycling and homeostasis maintenance. Reduction of the activity of the KCNQ4 channel has been associated with a genetic form of hearing loss, noise-induced hearing loss, and age-related hearing loss [6,7,8]; therefore, small compounds that activate KCNQ4, the so-called “channel openers”, have been developed as a strategy for the treatment of these hearing impairments [9]. In this review, we focused on the developmental status of KCNQ4 activators and compared their advantages and shortcomings in terms of their potential to be used for the specific activation of KCNQ4.

## 2. KCNQ Potassium Channels

The KCNQ family of voltage-gated potassium channels (Kv7) includes five members (Kv7.1–Kv7.5) that have important roles in the brain, heart, kidney, and inner ear [10]. In particular, KCNQ channels consist of a K^+^ channel pore-forming subunit (α-subunit) with six transmembrane domains (S1–S6) and a single pore-loop (P-loop), and two intracellular termini (Figure 1). Functional KCNQ channels are assembled in homo- or heterotetramer pore-forming subunits. The proteins have been shown to share between 30 and 65% amino acid identity, with particularly high homology in the transmembrane regions (Figure 1) [11,12]. The S4 transmembrane domain containing a regular distribution of positively charged amino acids acts as the voltage sensor, while the P-loop contains the K^+^ pore TxxTxGYG signature sequence (Figure 1). The length of the N-terminus, which is in the order of 100 amino acids, is similar between the five subtypes, whereas the length of the C-terminus varies greatly between the subtypes. All five proteins display a highly homologous region on their intracellular C-terminus termed “A-domain” (Figure 1) [11]. The high homology in critical residues, such as the voltage sensor domain and P-loot, has hindered the development of subtype-specific activators, including KCNQ4-specific activators.

All five Kv7 members map to a human disease locus. Mutations in KCNQ genes have been shown to cause inherited syndromic diseases. More specifically, mutations of *KCNQ1* are known to cause heart diseases, including long QT syndrome (MIM 192500), and Jervell and Lange-Nielsen syndrome (MIM 220400) in autosomal dominant and recessive manners, respectively [13,14,15]. Mutations of *KCNQ2* have been found to cause autosomal dominant benign familial neonatal seizures (MIM 121200) [16,17]. Mutations of *KCNQ3* have also been shown to cause autosomal dominant benign neonatal seizures (MIM 121201) [18], while mutations of *KCNQ4* result in autosomal dominant nonsyndromic hearing loss (DFNA2, MIM 600101) [6]. Finally, mutations of *KCNQ5* have been reported to cause autosomal dominant intellectual disability (MIM 617601) [19].

Among the 30 genes associated with autosomal dominant hearing loss, KCNQ4 is one of the most commonly mutated genes [20,21]. *KCNQ4* mutations explained 6.62% (19/287) in Japanese families with autosomal dominant nonsyndromic hearing loss and c.211delC was identified as a founder mutation in Japanese individuals, explaining 68.4% (13/19) among families with *KCNQ4* mutations [20]. In addition, in our Korean adult-onset hearing loss patient cohort (i.e., Yonsei University Hearing Loss or YUHL cohort) without noise exposure history, *KCNQ4* presented the highest prevalence for mutations (9/213 patients, unpublished). In particular, DFNA2 resulting from mutations in *KCNQ4* is characterized by progressive sensorineural hearing loss at all frequencies [6,22]. The progressive nature of DFNA2 is advantageous for treatment because it provides a wide therapeutic window if the causative mutations could be detected early. Since the first clinical report of a mutation of *KCNQ4* responsible for deafness in 1999 [6], over 40 pathogenic mutations have been identified in individuals with DFNA2 (www.deafnessvariationdatabase.org or www.hgmd.cf.ac.uk/ac/index.php, accessed on 31 December 2020) [21]. The mutation hotspots in *KCNQ4* associated with DFNA2 have been shown to be clustered around the pore region [21]. Variants in the pore region of KCNQ4 are known to be unresponsive to KCNQ activators, such as retigabine or zinc pyrithione [21,23,24]. In addition, it was found that among pore region variants, variants that result in almost null potassium activity did not respond to KCNQ activators, whereas variants with residual voltage-activated K^+^ currents could be activated by KCNQ activators [21]. In addition, variants occurring in the N- and C-terminal cytoplasmic termini had higher chances to be rescued by KCNQ4 activators [21,25]. However, the relationship between each mutation and drug responsiveness remains unclear.

## 3. Potassium Recycling and KCNQ4 in the Inner Ear

The inner ear of mammals contains two sensory organs, the cochlea and the vestibule, which are responsible for hearing and balance, respectively. The cochlea consists of three fluid-filled compartments with different ion compositions: (1) scala vestibuli; (2) scala media; and (3) scala tympani. The scala vestibule and scala tympani are filled with perilymph, whereas the scala media is filled with the endolymph, which has a high K^+^ concentration and a positive potential [26]. The mammalian cochlear contains two types of sensory cells with a bundle of actin-based stereocilia on their apical surface: (1) outer hair cells (OHCs), which amplify sound stimuli; and (2) inner hair cells (IHCs), which transmit sound stimuli to the central nervous system [27]. The sensory cells of cochlea are bathed in endolymph and a difference in K^+^ concentration is maintained between the endolymph and the sensory cells in the scala media. As K^+^ is the major charge carrier for the sensory transduction, its proper recycling is of great importance for the process of hearing. Briefly, K^+^ ions are secreted into the endolymph by the stria vascularis, enter the sensory OHCs through apical mechanosensitive K^+^ channels, probably including transmembrane channel like 1 (TMC1) and TMC2 [28], thereby triggering neurotransmission, and are released from these cells into the perilymph via basolateral K^+^ channels, including KCNQ4. Then, they migrate through supporting cells and fibrocytes toward the stria vascularis using a network of gap junctions [26]. Accordingly, K^+^ recycling genes were shown to be indispensable for the process of hearing, as evidenced by the fact that multiple mutations in these genes (*GJB1* (Cx32), *GJB2* (Cx26), *GJB3* (Cx31), *GJB4* (Cx30.3), *GJB6* (Cx30), *KCNE1*, *KCNQ1*, and *KCNQ4*) lead to both syndromic and nonsyndromic forms of hearing loss [6,14,15]. Moreover, mice deficient for SLC12A2, a Na^+^/2Cl^−^/K^+^ cotransporter, and KCNJ10 were reported to develop hearing loss due to collapsed endolymphatic spaces and the inability to generate an endocochlear potential, respectively [29,30].

In order for the cochlear to respond to the dynamic range and speed of sound, fast electromechanical amplification of sound and fast repolarization of the receptor potential by OHCs is required [31]. The K^+^ current that is known to dominate in OHCs is termed I_K,n_ [32]. Mechanoelectrical transducer channels have been shown to be opened by deflections of the hair cell bundles at cochlear regions of specific frequencies, causing an influx of K^+^, which would in turn lead to the depolarization of the membrane and contraction of OHCs by the motor protein prestin [33]. The fast amplification process depends on the capacitance of OHCs at resting membrane potential, which is determined by the conductance of OHCs maintained through the KCNQ4-mediated efflux current I_K,n_ [34]. The expression and current of KCNQ4 were reported to be detected prior to hearing onset along the entire basolateral membrane of OHCs in mice [34,35]; however, after the onset of hearing (postnatal day 12–14), its expression was redistributed and restricted to the basal pole [36]. This expression pattern was demonstrated to correlate with its function in extruding K^+^ ions [32,37,38]. Moreover, KCNQ4 is known to be also expressed in IHCs, spiral ganglion neurons, and several nuclei along the auditory pathway, for example, cochlear nuclei and inferior colliculus [37,39,40]. However, it remains controversial whether it is expressed in IHCs or spiral ganglions, and whether there is a tonotopy in the expression in IHCs [41,42]. Accordingly, *Kcnq4*^−/−^ mice were reported to exhibit progressive hearing loss, with OHCs slowly decreasing at a young age with increasing cell loss leading up to complete degeneration at the oldest ages [42,43]. Degeneration of IHCs, particularly at the basal turn was also observed, but only in the adult stage [30]. The loss of this important K^+^ channel in OHCs is known to result in a chronic depolarization, possibly increasing Ca^2+^ influx through voltage-gated Ca^2+^ channels and causing their subsequent degeneration due to chronic cellular stress [44].

When expressed alone in CHO cells, KCNQ4 displays a half-activation voltage of −19 mV and a slope constant of 10 mV. The activation onset has been found to be exponential, except at very positive voltages, displaying little or no inactivation [45]. In oocytes, the half-activation voltage has been demonstrated to be −10 mV, the slope constant 18 mV, and the activation slow, with a time constant of 600 msec at +40 mV [6].

## 4. Association of KCNQ4 and Noise-Induced Hearing Loss

Noise-induced hearing loss is estimated to affect 12% or more of the global population and has become a leading occupational health risk in developed countries [46]. The World Health Organization estimated that 1.1 billion young people worldwide are at risk of developing hearing loss due to noise exposure [47].

Genetic factors are also known to contribute to noise-induced hearing loss [48]. Therefore, the association of K^+^ recycling gene variants with susceptibility to noise has been examined. Van Laer et al. investigated the association of 35 single nucleotide polymorphisms (SNPs) in 10 genes including *GJB1*, *GJB2*, *GJB3*, *GJB4*, *GJB6*, *KCNJ10*, *KCNQ4*, *KCNQ1*, *KCNE1*, *KCNQ3*, and *SLC12A2* on 104 noise-susceptible and 114 noise-resistant individuals selected from a population of 1261 Swedish noise-exposed workers [7]. They found that three SNPs in *KCNE1*, one SNP in *KCNQ1*, and one SNP in *KCNQ4* were significantly associated with noise-induced hearing loss [7]. Another association study was performed in 119 noise-susceptible and 119 noise-resistant individuals selected from a population of 3860 Polish noise-exposed workers [49]. Pawelczyk et al. examined 99 SNPs in 10 K^+^ recycling genes and found a significant association of SNPs in 7 out of 10 genes (*KCNE1*, *KCNQ4*, *GJB1*, *GJB2*, *GJB4*, *KCNJ10*, and *KCNQ1*) [49]. Two SNPs, the rs2070358 G allele in *KCNE1* and rs34287852 G allele (c.1365T>G, p.H455Q) in *KCNQ4*, were reported to be significant in both populations, with the rs2070358 G allele increasing the susceptibility to noise-induced hearing loss [7,49]. Interestingly, the rs34287852 G allele in *KCNQ4* exhibited the opposite effect in these two populations, as it was found to decrease the risk of developing noise-induced hearing loss in the Swedish population, but increased the susceptibility of noise-induced hearing loss in the Polish population [7,49]. This discrepancy could be theoretically explained by differences in allele frequency or linkage disequilibrium patterns in both populations, slightly different selection procedures applied in both studies, the influence of various environmental factors, or, finally, by a false positive association with noise-induced hearing loss of the rs34287852 G allele of *KCNQ4* [50]. The allele frequency of rs34287852 is 0.1763 in total; however, ethnic differences exist. It has been shown to be more common in Europeans with an allele frequency of over 0.15, whereas it is less than 0.07 in eastern Asians and Africans. This SNP is known to result in a missense change (p.H455Q). Jung et al. showed that the activity of the K^+^ channel of this variant was not different from that of wild-type KCNQ4 and increased to a level similar to wild-type KCNQ4 following administration of retigabine [21]. Last, Guo et al. examined three SNPs (rs709688, rs2769256 and rs4660468) for an association with noise-induced hearing loss on 571 cases and 639 normal controls selected from about 2700 noise-exposed Chinese workers [51]. They found that one synonymous SNP, the rs4660468 T allele, was significant, conferring a higher risk of noise-induced hearing loss [51].

Acoustic noise exposure has been suggested to decrease the functionality of KCNQ4 on the surface membrane, thereby playing a pivotal role in noise-induced hearing loss. Loss of KCNQ4 on the membranes of OHCs in cochlear regions of high frequency was reported to precede the loss of OHCs in mouse models [44,52]. Likewise, KCNQ4 was also reported to be lost from the surface membrane of OHCs in cochlear regions of low frequency following exposure to low frequency noise [53]. Therefore, KCNQ4 was assumed to protect OHCs from Ca^2+^ overload triggered by noise exposure [42,53].

## 5. KCNQ4 Activators

As we discussed, the common molecular basis of DFNA2 and noise-induced hearing loss is the reduction of the activity of KCNQ4 in OHCs, resulting from either mutations or noise exposure; therefore, restoration of the activity of KCNQ4 is a logical strategy for the treatment and prevention of these conditions. To this end, a number of synthetic compounds that potentiate KCNQ channels have been developed to treat diseases resulting from neuronal hyperexcitability, such as epilepsy and neuropathic pain [54]. Some of these chemicals have been examined for their ability to activate KCNQ4. In addition, efforts have been made to develop compounds specific to KCNQ4 over other KCNQ channels.

### 5.1. Retigabine

Retigabine, also known as ezogabine, is a first-in-class drug for the treatment of epilepsy, approved by the US Food and Drug Administration [55,56,57,58]. Retigabine has been the most characterized activator of KCNQ channels and has been shown to potentiate KCNQ2, KCNQ3, KCNQ4, and KCNQ5, without activating KCNQ1, thereby avoiding potential cardiac effects [56,59,60,61]. Due to its broad effect on various subtypes of KCNQ channels, retigabine is also utilized as an antidepressant [62], an antihypertensive [63], an analgesic [64], an anxiolytic [65], and even as an antimanic [66]. However, its administration has been associated with side-effects, such as retinal pigmentation, urinary retention, and skin discoloration [58,67]. Although it is known to serve as an activator, it has also been shown to inhibit KCNQ channels at positive potentials [68]. In addition, retigabine is known to act on other channels, including g-aminobutyric acid receptor channels [69].

Retigabine has an effective concentration for half-maximum response (EC_50_) of 1.4 μM at −30 mV [70] and 3.7 μM at 0 mV [23] for KCNQ4 in vitro, with 10 μM of retigabine increasing the native I_k,n_ currents by 1.4-fold at −60 mV to 2.2-fold at −110 mV [23]. Moreover, retigabine has been shown to shift voltages of activation to hyperpolarized potentials [23]. Li et al. determined the structures of KCNQ4 and its complex with retigabine using cryoelectron microscopy [71]. Four retigabine molecules were demonstrated to bind to one KCNQ4 tetramer, with each retigabine residing in a single hydrophobic fenestration site in the middle of the membrane. Retigabine contains three major functional groups: the fluorophenyl group, the middle phenyl ring, and the carbamate group (Figure 2). The Trp242, Phe246, Leu249, Leu305, Leu306, Ser309, Phe310, Phe311, Pro314, and Leu318 residues of KCNQ4 have been shown to be involved in the binding of retigabine (Figure 2) [71]. Using systematic mutagenesis studies, it was identified that the tryptophan residue (Trp242 in KCNQ4) in S5 was crucial for the activity of retigabine and was further shown that it is conserved from KCNQ2 to KCNQ5 [61,72], but replaced by Leu266 in KCNQ1 (Figure 1) [71]. Both the side chain of Ser309 and the carbonyl oxygen of Leu305 can form hydrogen bonds with the amino group from the carbamate group of retigabine [71]. In addition, both the side chain of Ser309 and the carbonyl oxygen of Phe311 can form hydrogen bonds with the amino group from the middle phenyl ring of retigabine [71]. The side chain of Trp242 and the aromatic ring of Phe110 have been reported to interact with the carbonyl oxygen from the carbamate group and the fluorine atom of retigabine, respectively [71]. In particular, KCNQ4 was shown to be modulated by phosphatidylinositol 4,5-bisphosphate (PIP_2_) which is known to activate KCNQ channels by coupling voltage-sensing domains and the central pore domain [73,74]; a single molecule of PIP_2_ inserts its head group into a cavity within each voltage-sensing domain [71].

Even though retigabine has been extensively studied and shown to prevent salicylate-induced ototoxicity in rats [75], its use against hearing loss has been limited due to the potential side-effects resulting from its broad action on KCNQ channels.

### 5.2. Retigabine Derivatives

Wang et al. reported several compounds that were made by modifying retigabine and showed better selectivity for KCNQ4 and KCNQ5 [76]. For instance, a N-1/3 substitution resulted in improved specificity for KCNQ4 and KCNQ5 compared with naïve retigabine [76]. Especially, 10 g of one of those derivatives showed the best potency for KCNQ4 and KCNQ5 with EC_50_ values of 0.78 and 1.68 μM, respectively, and had a minimal effect on homomeric KCNQ2 [76]. More specifically, 10 μM of 10 g of this compound was reported to increase the currents of KCNQ4 and KCNQ5 by 6.4- and 4.6-fold, respectively [76]. Further modification of this compound may lead to a drug with better specificity for KCNQ4 over KCNQ5. This study also demonstrated that alteration of chemical subunits from currently available KCNQ channel activators might serve as a promising platform for the discovery of novel compounds targeting KCNQ4 with higher specificity [77].

NS15370, which was developed as a chemical retigabine analog with higher potency [78], was shown to induce a shift in the voltage-dependence of activation, enhancing KCNQ4-mediated currents at potentials negative to 0 mV, but suppressing them at more positive membrane potentials [79].

### 5.3. Zinc Pyrithione

Zinc pyrithione (ZnPy), which is widely used for dandruff and psoriasis [80], has been shown to activate KCNQ channels, except KCNQ3 and KCNQ5 [81]. Accordingly, 10 μM ZnPy increased KCNQ4-mediated currents by 76.1-fold at -30 mV and 23.5-fold at +50 mV [81] and potentiated the native I_k,n_ currents when combined with retigabine [23]. It should be mentioned that ZnPy is unique in that it increases the open probability of KCNQ2 and KCNQ4 in addition to inducing a hyperpolarizing shift in the voltage dependence of activation and increasing the current amplitude [81]. However, the activity of ZnPy does not depend on the tryptophan residue in S5, which is different from retigabine, but on the interaction with the pore region (Figure 2) [82]. Furthermore, both Zn^2+^ and pyrithione were demonstrated to be essential for activity, with the potency depending on the proper stoichiometry of 1:2 zinc-to-pyrithione [81].

### 5.4. Maxipost

Maxipost, formerly known as BMS-204352, was identified as a potent opener of calcium-activated maxi-K channels (BK channels) used for the control of convulsion and stroke [83]. Maxipost is known to be a potent activator of KCNQ channels with an EC_50_ of 2.4 μM for KCNQ4 at −30 mV [65,70], exhibiting protective effects against peripheral salicylate ototoxicity [75], and reported to abolish behavioral evidence of tinnitus [84]. Maxipost has been found to shift the voltages of activation to hyperpolarized potentials, which are dependent on the tryptophan residue in S5 (Figure 2) [65,70,85], but failed to potentiate native I_k,n_; therefore, its application for hearing loss has been limited [23].

### 5.5. Acrylamide (S)-1

Acrylamide (S)-1 was synthesized as an orally bioavailable KCNQ2 activator for the control of migraines [86]. Despite its development for KCNQ4, acrylamide S-(1) was shown to be preferentially specific for KCNQ4 and KCNQ5 [85]. The EC_50_ for KCNQ4 determined for acrylamide S-(1) in *Xenopus* oocytes was 10.4 μM at 0 mV, with 100 μM acrylamide (S)-1 leading to a 20-fold enhancement of current amplitude [85]. The effect of acrylamide (S)-1 on KCNQ4 was reported to be potentiated across all voltage levels [85], whereas its effect on KCNQ2 and KCNQ3 was shown to be voltage-dependent [87]. Another study revealed that the hyperpolarizing shift induced by acrylamide (S)-1 depended on the conserved tryptophan residue in S5 (Figure 2) [79], which is also required for retigabine and maxipost, suggesting that these three compounds might exert their effects on KCNQ4 in a similar manner.

### 5.6. Other KCNQ4 Activators

There have been additional compounds reported to activate KCNQ4. AaTXKβ_(2–64)_, a toxin isolated from the North African scorpion, was shown to specifically activate KCNQ3 and KCNQ4, without affecting KCNQ1 and KCNQ2 [88]. This peptide had an EC_50_ of 58 μg/mL at 0 mV, inducing a hyperpolarizing shift, and increasing KCNQ4 currents by 2-fold at 0 mV [88]. Fasudil, a rho-associated kinase inhibitor and a vasodilator, was found to potentiate KCNQ4 and KCNQ4/5 with EC_50_ values of 12.9 and 15.7 μM, at −30 mV, respectively, without affecting KCN2 and KCNQ2/3 [89]. Fasudil shifted the voltage-dependent activation curve in a more negative direction, for which the Val248 in S5 and Ile308 in the S6 segment of KCNQ4 were required [89]. ML213, identified by a high throughput fluorescent screen of the NIH Molecular Libraries Small Molecule Repository and structure–activity relationship, was initially characterized as specific for KCNQ2 and KCNQ4, with a 80-fold selectivity over other KCNQ channels [90]. The EC_50_ of ML213 for KCNQ4 was shown to be 1.8 μM at −10 mV, with ML213 inducing a hyperpolarizing shift, which was reported to be dependent on the crucial tryptophan residue in S5 and a 2-fold increase in currents following administration of 10 μM of the drug (Figure 2) [79]. In the cases of AaTXKβ_(2–64)_, fasudil, and ML213, it is necessary to examine whether these drugs could induce the native I_k,n_ currents of OHCs.

Through the Cortellis Drugs Discovery Intelligence™ service by Clarivate, we found that several pharmaceutical companies and universities are currently developing KCNQ4 activators. Most of them are being explored for their potential to treat neurological disorders, including epilepsy, pain, migraines, and many more conditions, and are currently in the phase of biological or preclinical testing. Acousia Therapeutics, which is a biotech company aiming for the development of small-molecule drugs for sensory neuronal hearing loss, has eight compounds targeting KCNQ4 in its pipeline.

## 6. Conclusions and Future Directions

Understanding of the function, structure, physiology, pharmacology, and genetics of KCNQ4 has indicated that KCNQ4 holds great promise for the discovery and development of drugs useful in genetic, age-related, and noise-induced hearing loss. Due to the growing prevalence and socioeconomic burden of hearing loss, the demand for drugs aimed in controlling these conditions have been increasing.

Application of available KCNQ4 activators has been currently restricted due to the lack of subtype specificity, especially for KCNQ2–KCNQ5, as well as due to their insufficient in vivo efficacy. As a member of voltage-gated K^+^ channels, KCNQ4 shares structural and amino acid similarities with other KCNQ channels, which hinders the development of KCNQ4-specific drugs. As other KCNQ channels are involved in the physiology of diverse tissues and may also be associated with diseases, lack of specificity would lead to unwanted side-effects. As such, the selectivity for KCNQ4 could be achieved by the further refinement of existing compounds, as shown previously [76]. Based on the delineation of the crucial residues of the specific binding of retigabine with KCNQ4 and the molecular details of its activation [71], the discovery and optimization of related chemicals would be accelerated. Knowledge of the structure of KCNQ4 should also enable the design of compounds that induce conformational changes with an outcome similar to that normally caused by membrane depolarization. Moreover, simultaneous application of two activators with distinct modes of action may result in synergistic effects and reduced side-effects. In addition, considering the anatomical characteristic of the inner ear, as an isolated organ, using local administration as the effective delivery route might serve as both the main point of concern in the pharmacological development process, as well as an opportunity for reducing side-effects. Limited volume of therapeutic materials for administration in the inner ear without overloading the cochlea might require high efficacy and concentrations in order to reach the maximal impacts on hearing rescue.

Although some compounds could activate KCNQ4 in vitro, none of them were potent in inducing the native I_k,n_ currents in OHCs [23]. The molecular basis for this discrepancy is not yet clear and OHC-specific modifiers were suggested [23]; however, more thorough studies regarding this aspect are required because many compounds have not been properly evaluated in this regard. The efficacy of KCNQ4 activators needs to be examined not only in OHCs in explants, but also in Kcnq4 mouse models. Besides, KCNQ4 activators would not be effective in Kcnq4^−/−^ mice as there is no KCNQ4 to be activated. Similarly, they would not be effective in knock-in mice harboring the p.G286S (c.856G>A) pore region mutation, which results in unresponsiveness to KCNQ activators [21,42]. Therefore, it is necessary to generate additional Kcnq4 mice that would harbor pore region mutations with residual activity of K^+^ channels or variants in the two cytoplasmic termini of KCNQ4 to investigate the in vivo efficacy of channel openers. In addition, the effectiveness of KCNQ4 activators needs to be examined in mouse models of noise-hearing loss.

More importantly, KCNQ4 activators should be validated in clinical trials, as there is no ongoing clinical trial targeting hearing loss by KCNQ4 activators currently. Target population, such as individuals with genetic hearing loss or noise-induced hearing, should be carefully selected. Both the therapeutic window and convenient application methods are important issues to be discussed in clinical trials. Drug repurposing and optimization for applicable specific *KCNQ4* mutation might also be an option for clinical application of KCNQ4 activators in deafness treatment with advantages of reducing the cost and shortening the time when compared to de novo drug discovery. In addition, the half-life duration and bioavailability of drugs targeting the activation of KCNQ4 would also be required to satisfy clinical degrees.

## Figures and Tables

**Figure 1 ijms-22-02510-f001:**
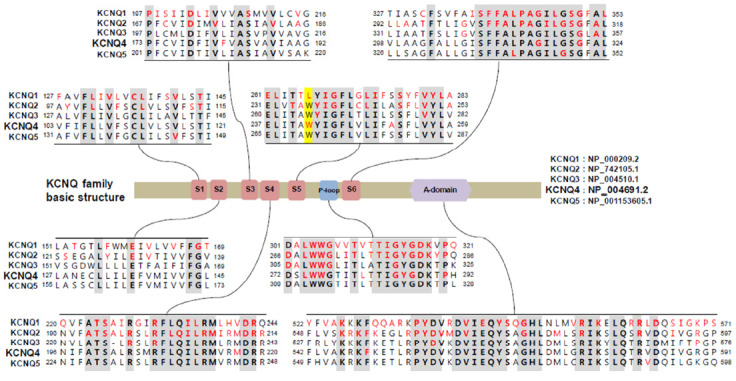
Comparative sequence analysis among voltage-gated channel subfamily q (KCNQ) family genes and mutational spectrum in protein sequences. Conserved sequences are presented in bold characters. Amino acids affected by pathogenic mutations reported in associated Mendelian diseases were collected from the HGMD and ClinVar databases and are presented in red characters. KCNQ2–4 share the tryptophan residue (Trp242 in KCNQ4, highlighted in yellow), which is critical for the activity of several KCNQ activators, including retigabine; however, KCNQ1 has a leucine at this position, and not a tryptophan.

**Figure 2 ijms-22-02510-f002:**
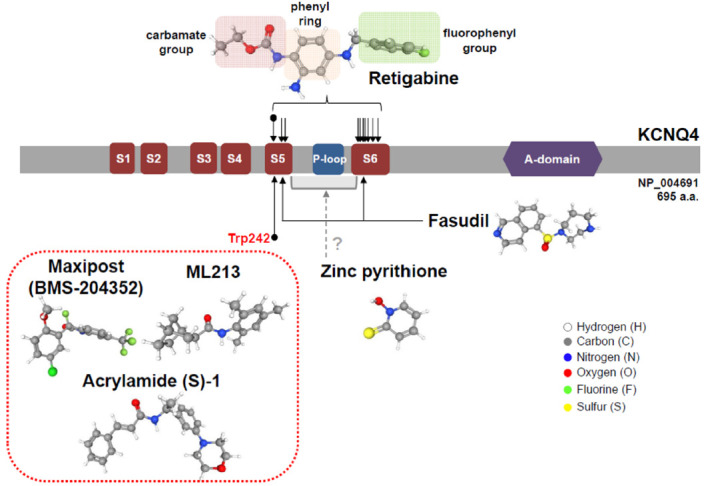
Pharmacological action sites of KCNQ4 activators along functional domains. Critical amino acid residues for the action of KCNQ4 activators are mostly located in the S5 and S6 regions, including the crucial site Trp242. Retigabine, maxipost, acrylamide (S)-1, and ML213 require this tryptophan residue for their activity.

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
