# Peer review of "Activation of KCNQ4 as a Therapeutic Strategy to Treat Hearing Loss"

_ijms, 2021, doi:10.3390/ijms22052510_

Round 1
Reviewer 1 Report
this article discusses an interesting aspect of research to treat hearing loss. the important function of the KCNQ potassium channel has been extensively analyzed and is comprehensive.
some parts to be further investigated:
-paragraph 4: in order to compare the results it would be useful to report some details of the studies examined such as age of the population studied, level of hearing loss, type of noise exposure
- continue research on KCNQ4 activators with clinical trials to evaluate the actual potential for clinical use and study side effects
Reviewer 2 Report
REVIEW
In an article by John Hoon Rim , Jae Young Choi, Jinsei Jung, and Heon Yung Gee “Activation of KCNQ4 as a therapeutic strategy to treat hearing 2 loss”.
The article by John Hoon Rim and co-authors is review by updated the current concept of the physiological role of KCNQ4 in the inner ear and the 17 pathologic mechanism underlying the role of KCNQ4 variants with regard to hearing loss.
Major comment
No major comments, this is an interesting review.
Minor comments
1) Nomenclature of genes needs for correction:
- The names of human genes should be written in capital letters, in italics;
- Non-human: in capital letters, in italic;
2) Nomenclature of mutations are needs for correction in one style:
- In nucleotids;
- In amino acids.
Recommendation
Accept after revision of minor comments
Sincerely yours,
Dr Nikolay A. Barashkov
Head of Laboratory of Molecular Genetics,
Yakut Scientific Centre of Complex Medical Problems,
677010, Sergelyahsoe shosse 4, +7(4112) 321981,
Sakha Republic, Russia
Reviewer 3 Report
The review by Hoon Rim and coauthors titled: "Activation of KCNQ4 as a therapeutic strategy to treat hearing loss" provides an overview of whether hearing loss due to ear channel openings could be treated with KCNQ4 as a target. The promising title leads the reader to expect that there may be possible therapies as was suggested 10 years earlier.
Why should one read the review article? Is it informative and introduces the topic well, here for example a statement about non-syndromic genetic deafness, age-related hearing loss is missing. One would also like to know more about the contribution of KCNQ4 mutations to the different deafness disorders.
It should be mentioned here that repurposing is a very good strategy to avoid the high hurdles of approval. However, the problem is that each mutation seems to need an individual pharmacology? This should be more specified in the article.
KCNQ4 is also target for numerous blockers like bepridil uv.a, this is not mentioned at all. What is the actual physiological problem of potassium homeostasis? This could be made clearer.
Do OHC cells lose too little potassium in the KCNQ4 channels? Mutated KCNQ4 channels seem to require higher voltages. This may be bypassed after cochlear implantation, for example, because higher voltage peaks stimulate residual hearing.
Would one expect higher potassium levels in KCNQ4 mutant cells? What is the role of anion homeostasis here?
Or do multiple mutations cause the cells to become potassium-depleted after all and therefore unable to reach or compensate for the reversal potentials? What is the role of KCNQ heteromers?
The reference list misses some literature references:
Ramzan M et al., 2019; Mehregan H et al., 2019; Hosseinzadeh Z et al., 2013; Strutz-Seebohm N et al., 2006; Wu et al., 2010; Huang et al., 2017; Su et al., 2007; Gao et al., 2013; Wang et al., 2014; Van Eyken E et al. 2006;
Round 2
Reviewer 1 Report
The changes made add interesting details to the article, probably no further information is available on the populations examined, although considered important as per the previous review. We believe that the importance of further research into therapeutic possibilities is to be encouraged, even though no studies appear to be active at present.